# Numerical and Experimental Assessment of the Effect of Residual Stresses on the Fatigue Strength of an Aircraft Blade

**DOI:** 10.3390/ma14185279

**Published:** 2021-09-14

**Authors:** Arkadiusz Bednarz, Wojciech Zbigniew Misiolek

**Affiliations:** 1The Faculty of Mechanical Engineering and Aeronautics, Rzeszow University of Technology, Al. Powstancow Warszawy 12, 35-959 Rzeszów, Poland; 2The Loewy Institute, Lehigh University, Bethlehem, PA 18015, USA; wzm2@lehigh.edu

**Keywords:** compressor blade, fatigue life, shot-peening, x-Ray diffraction, residual stresses

## Abstract

The work presents the results of numerical fatigue analysis of a turbine engine compressor blade, taking into account the values of initial stresses resulting from surface treatment-shot-peening. The values of the residual stresses were estimated experimentally using X-ray diffraction. The paper specifies the values of the residual stresses on both sides of the blade and their reduction due to cutting through the blade-relaxation. The obtained values of the residual stresses were used as initial stresses in the numerical fatigue analysis of the damaged compressor blade, which was subjected to resonant vibrations of known amplitude. Numerical fatigue ε-N life analysis was based on several fatigue material models: Manson’s, Mitchell’s, Baumel-Seeger’s, Muralidharan-Manson’s, Ong’s, Roessle-Fatemi’s, and Median’s, and also on the three models of cyclic hardening: Manson’s, Xianxin’s, and Fatemi’s. Because of this approach, it was possible to determine the relationship between the selection of the fatigue material ε-N model and the cyclic hardening model on the results of the numerical fatigue analysis. Additionally, the calculated results were compared with the results of experimental research, which allowed for a substantive evaluation of the obtained results. These results are of great scientific and practical importance. The problem of determining the fatigue life of blades with defects operating under resonance vibrations is one of the original tasks in the field of fracture mechanics and experimental mechanics. The results obtained are of great importance in the aviation industry and can be used during engine maintenance and inspections to assess the suitability of blades with defects in terms of the needs of further work. This aspect of engineering maintenance is of great importance from the aircraft safety point of view.

## 1. Introduction

Compressor blades are included in the group of critical elements of the aircraft engine. The complexity of the load system acting on the compressor blades as well as many phenomena taking place during the operation of the turbine does contribute to the reduction of the blade’s fatigue durability. The basic loads are the gasodynamic forces inducing bending force and the centrifugal force that causes stretching of the blade [1]. One of the factors affecting the reduction of blade life is the possibility of operational damage [1,2,3]. Compressor blades rotating while the engine is running create a negative pressure in the zone in front of the air inlet. The sucked air stream can lift loose elements from the landing surface. These elements, when entering the engine inlet, often collide with the compressor blades [4]. Compressors’ first-stage blades, due to their location, are most vulnerable to collision with foreign objects.

Hard objects (e.g., stones, sand) could cause collisions with the rotating blades of the compressor and result in damage in the form of cracks and notches [3,4]. Depending on the shape, hardness, and size of the elements sucked into a turbine engine, different types of damage can be created, which have a significant impact on the fatigue life of the blade. Observations of damaged engines indicate that the leading edge of the blade is primarily exposed to mechanical damage (Figure 1). Occasionally, damage also occurs at the trailing edge of the blade.

The work of the blade (containing a notch formed as a result of a collision with a hard object) under resonance conditions may cause rapid initiation of cracks and fatigue damage, resulting in damage and immobilization of the engine [4,5].

The problem of operational damage of the compressor blades has been the subject of research by many authors [2,3,4,5]. In these studies, the attention was focused mainly on explaining the causes of cracks. The factor that is important from the point of view of the operation of aircraft engines, which is the speed of propagating cracks in the compressor blade, is rarely analyzed in the scientific literature. In many cases, the results of experimental studies were enriched by the results of complex numerical analyses showing the distribution of stresses in the notch zone. The problem of initiation and propagation of cracks as well as analysis of fatigue life of aircraft engine components is discussed in the literature [5,6,7,8,9]. In these studies, the problems related directly to the issues of operational damage were considered, but the impact of the notch geometry, its location, and the method of defect formation on the fatigue life of the blade was not analyzed in detail.

An important aspect of the fatigue tests and their investigations is the determination of the working conditions and the assessment of the geometry and surface treatment of the analyzed object [10,11]. Knowledge of the stress state during the service as well as in the static (unloaded) state, in which the presence of the residual stresses is a result of earlier surface treatment. There are several methods for determining the value of the residual stresses, one of the most accurate and the most sensitive is X-ray diffraction [11,12,13,14]. Knowledge of the residual stresses can be included in numerical calculations to obtain results closer to the actual fatigue life results [13,14,15,16]. Such recognition of the topic may allow increasing the safety of air transport.

The main purpose of this study was to determine the fatigue life of a notched compressor blade, taking into account the values of the residual stresses resulting from shot-peening. The value of the residual stresses was determined using X-ray diffraction. Additionally, based on the literature fatigue models for the material used [17,18,19,20,21] and the models of cyclic hardening [21,22,23,24,25,26], the fatigue life of the blade was numerically estimated. The results of the numerical fatigue analysis were compared with the experimental results presented in the publication [27]. As the result of this approach, the level of residual stresses in the blade was determined, as well as their impact on the fatigue life. An additional result was information on the impact of the fatigue ε-N material model and the cyclic hardening model on blade performance. The obtained results may contribute to a better understanding of fatigue phenomena in objects with complex load conditions and complex geometry. The presented results may also be useful in eventual future forensic investigations as a source of information on the impact of the manufacturing process (like peening) of an aircraft engine compressor blade on its fatigue strength.

## 2. Examination

In the first step of the research, the level of residual stresses in the compressor blade was determined. X-ray diffraction was used as a technique to determine the level of residual stresses. This residual stress was interpreted as the initial stress in numerical fatigue analysis.

### 2.1. Experimental Object

The test object was the blade from the first stage vane of the PZL-10W turbine engine (Pratt&Whitney AeroPower Rzeszow, Rzeszow, Poland) (Figure 2). The weight of the examined blade was 15.85 g. The blade of the first stage compressor was made of EI-961 steel (GOST: 13Kh12N2V2MF). This steel has a chromium content of around 11% [28,29]. Other alloying elements (in order of the highest to the smallest percentage) are tungsten, nickel, silicon, molybdenum, carbon, vanadium, phosphorus, and sulfur. The density of this alloy is 7850 kg/m^3^. EI-961 alloy steel has the following strength properties (at 20 °C):Ultimate Tensile Strength: UTS =1050 MPa;Yield Strength: YS = 850 MPa;Young Modulus: E = 200 GPa;Poisson ratio: υ = 0.3.

### 2.2. Residual Stress Measurements

Two blades were used in the research (Figure 2b). The residual stress was measured by an X-ray diffraction scanner (Figure 3). In both cases, the initial stresses were determined/measured on both surfaces of the blade. An example of a measurement point is presented in Figure 4. The obtained results are summarized in Table 1. The study used the Micro µ-X360s X-ray pre-stress analyzer by Pulstec^®^ (Pulstec Industrial Co., Ltd., Hamamatsu-City, Japan) (Figure 3).

The results of the tests showed that on the concave side of the blade, the lowest stresses of −713 MPa were located next to the foot of the blade. Shifting the measurement point to the height of h = 3 mm resulted in a change of residual stresses by about 50 MPa (to the value −659 MPa). At the same point, on the opposite side of the blade (convex side), the initial stresses of −230 MPa were observed. Initial stresses of about −470 MPa were observed at the tip of the blade.

The test results showed that on the surface of the blade there are negative pre-stressed areas (compressive stresses) of very high values. Such high values of compressive stresses prove that the blade was subjected to surface treatment such as peening.

The difference in the value of the initial stresses occurring on the concave and convex sides of the blade, at a 3 mm distance from the root, is about 430 MPa, which proves the different intensity of peening of the above-mentioned zones. The distribution of the initial stresses along the height of the blade is shown in Figure 5.

In addition to the measurement along with the blade height, the residual stresses were measured in a cross-section of 3 mm from the blade root. The measurement was made at six points–three on the inner and three on the outer side of the blade. The exact location of the measurement points is presented in Figure 6, and the measurement results are presented in Table 2.

It was found that (See Table 2) the highest value of residual stresses occurred on the inner side of the blade, in its central position (2) (−659 MPa), while the lowest value occurred on the outer side of the blade, at the trailing edge (4) (−197 MPa). From the viewpoint of the fatigue calculations related to the damage at the leading edge, the level of residual stresses in this part of the blade is very important. On the inner side, they amounted to −478 MPa, and on the outer, −213 MPa (more than two times lower than the stress value on the inner side of the blade). For the sake of simplicity, it was assumed that the average value of the residual stresses on the inner of the blade was −464 MPa, and on the outside was −213 MPa. As shown by experimental fatigue tests [27], the fatigue crack in the compressor blade propagates much faster on the inside of the blade, so the average value (σ_res_ = −464 MPa) was used in the numerical tests as a value of residual-initial stresses.

After carrying out the necessary measurements, the blade was cut into slices to perform an additional measurement of the residual stresses on their edges. As a result of the performed measurements, it was determined that the value of the residual stresses decreased for the point on the inside of the blade, at a height of 3 mm, from −659 MPa to −386 MPa (by 273 MPa). The observed reduction in residual stresses value was above 41%.

## 3. Fatigue Life Assessment

The next step in the performed research was to determine the stress level and prepare the ε-N fatigue models (Manson-Coffin-Basquin equation) [17,18,19,20,30] and models of cyclic hardening (for the Ramberg-Osgood equation) [23,31]. As part of the previous tests [30,31], it was determined that in the case of resonant vibrations of the compressor blade of the PZL-10W engine with a geometric notch with a depth of 0.5 mm located at a height of 3 mm from the blade root, with a known vibration amplitude A = 1.8 mm, the equivalent (Von-Mises) stress reached the highest value in the bottom of the notch, and it amounted to σ_eqv_ = 987 MPa. The maximum principal stresses σ_1max_ occurred at the same location and were 184 MPa higher [30]. The value of maximum principal stress σ_1max_ = 1171 MPa was used in later calculations as the basic quantity describing the blade load. The equations used in the calculations were presented in Section 3.2.

Previous studies [32] showed that in the discussed cross-section there was plasticization of the material, which was caused by peening. Unfortunately, due to the lack of information on the specifics of this surface treatment, it was not possible to model this phenomenon numerically.

### 3.1. Numerical Models of the Material Properties

To prepare a wide range of numerical tests, based on the available literature, 8 fatigue material models were selected for the Manson-Coffin-Basquin equation, based on the numerical fatigue ε-N analysis, which was carried out. The following fatigue models were used in the analysis: Manson’s, 4-point Manson’s, Mitchell’s, Muralidharan-Manson’s, Baumel-Seeger’s, Ong’s, Roessle-Fatemi’s, and Median’s [17,18,19,20,30,31]. Detailed data on fatigue models were included in Appendix A.

The determining quantities needed for fatigue calculations based on the Manson-Coffin-Basquin [30] Equation (1), i.e., fatigue strength coefficient (σ’_f_), fatigue ductility coefficient (ε’_f_), fatigue strength exponent (b), and fatigue ductility exponent (c), were estimated based on material data and static tensile test. The calculated values are summarized in Table 3 and presented in Figure 7.
(1)$$\varepsilon_{C}=\frac{\sigma \prime_{f}}{E}\cdot Nf^{b}+\varepsilon \prime_{f}\cdot Nf^{c},$$
(2)$$\varepsilon_{C}=\frac{\sigma }{E}+2{\left(\frac{\sigma }{2K^{\prime }}\right)}^{1/n^{\prime }},$$
where:σ′_f_–fatigue strength coefficient, *MPa*ε′_f_–fatigue ductility coefficientb–fatigue strength exponentc–fatigue ductility exponentNf–number of cycles to failure$\varepsilon_{C}$–total strain amplitudeσ–stress, *MPa*E–Young Modulus, *GPa*$K^{\prime }\;$–cyclic strength coefficient, *MPa*$n^{\prime }$–cyclic strain hardening exponent

Since the numerical strength analysis of a vibrating blade in the resonance state with an amplitude of 1.8 mm was performed based on a linear-elastic model, it is necessary to use a model taking into account the cyclic hardening (from the Ramberg-Osgood Equation (2)), i.e., cyclic strength coefficient (K’) and cyclic strain hardening exponent (n’). The Manson’s, Fatemi’s, and Xianxin’s models were selected for the analysis [14,15,16,31]. Some models depend on the fatigue model of the material. Detailed data on cyclic hardening models were included in Appendix A. The estimated values for calculations of the cyclical hardening are summarized in Table 4.

The presented fatigue models (Table 3) together with the models of cyclic hardening (Table 4) were used in later fatigue calculations. Fatigue analysis was performed based on the average value of the residual stresses on the inner side of the blade, at a height of 3 mm from the blade root (σ_res_ = −464 MPa). The second value used in the fatigue calculations was the maximum value of the principal stresses σ_1max_ = 1171 MPa, which was taken from the numerical strength analysis presented in the literature [30].

### 3.2. Algorithm of the Fatigue Life Assessment

Due to the complexity of the numerical strength analysis and hardware limitations facing the Finite Element Method (FEM) analysis, which could take into account the peeling process and resonance vibrations, it was decided to prepare a proprietary algorithm for estimating low-cycle fatigue life based on the following assumptions: a pendular sinusoidal load cycle was assumed, the principle of superposition was used (stresses add and subtract), the Ramberg-Osgood model of cyclic hardening described the hardening of the blade material very well, the fracture initiated at the edge of the geomatic notch on the concave side of the blade and the relaxation resulting from the removal treatment (machining) being the source of the notch was not taken into account. Based on the above assumptions and known equations, an algorithm for fatigue life estimation was prepared (Figure 8). The presented algorithm was prepared with the use of commercial Matlab^®^ software (R2021a).

The basic construction idea of the algorithm is the determination (with the assumed resolution) of the Manson-Coffin-Basquin (1) curve for a given fatigue model of the material and the conversion of stresses into strains using the Ramberg-Osgood Equation (2). The model assumes that a linear course is observed between two consecutive points on the Manson-Coffin-Basquin curve. Based on this approach, it is possible to use Tales’ theorem in fatigue life estimation. The very number of cycles to crack initiation was calculated, as mentioned above, based on the known deformations from the FEM numerical analysis (blades in the resonance vibrations) and the theorem on the similarity of triangles. Additionally, the proposed algorithm implemented the “for” loop several times to check all material configuration cases (24 cases in total). The fatigue material data and the corresponding data for the cyclic hardening model were imported from a previously prepared external file.

The work of the presented (Figure 8) algorithm begins with reading the values of the principial stresses and estimated residual stresses, for which further calculations will be carried out. Then, the fatigue data of all material models are loaded. In the next step, the stress for fatigue analysis is calculated based on the superposition principle. For the fatigue data entered, the algorithm computes the entire Manson-Coffin-Basquin curve (for every material configuration). Then, the algorithm converts the previously obtained stress value into a total deformation (elasto-plastic). After determining the deformation value, the algorithm searches the previously created Manson-Coffin-Basquin (MCB) fatigue curve and determines the area in which the durability will be estimated. In the next step, based on Thales’ theorem (and the assumption of a linear change between two consecutive calculation points from the MCB diagram), the algorithm estimates the fatigue life. These activities are performed in a loop for all material configurations (fatigue and hardening models). In the last step, the algorithm displays the estimated fatigue life.

The presented algorithm (Figure 8) made it possible to quickly estimate the number of cycles to crack initiation, under the assumed stress state. The obtained results for all 24 cases of material configurations are summarized in Table 5. In the case of the Manson hardening model, the highest durability occurred for the Manson fatigue model (7.4 × 10^3^), and the lowest for the 4-point Manson model. For the average model, the estimated durability was nearly 1000 load cycles. In turn, in the case of Ong’s fatigue model (and Fatiemi’s hardening model), the number of cycles to crack initiation was as high as 401 × 10^3^. These results are much higher than all the others calculations (more than 4 times greater than the second-highest result for a given hardening model-the Mitchell fatigue model). Interestingly, the smallest scatter in the results was observed for the model of cyclic hardening according to Xianxin. The highest value (1.61 × 10^3^) occurred for the Manson fatigue model and the lowest for the Mitchell model (0.13 × 10^3^). The smallest result was only 12 times smaller than the largest one. The value for the average model, in the case of the Xianxin cyclic hardening model, was 0.81 × 10^3^ and was slightly lower than the same fatigue model with hardening according to the Manson model.

### 3.3. Comparison of Numerical and Experimental Results

To correctly assess the results of the numerical fatigue analysis presented in Table 5, it was necessary to relate the obtained results to the number of cycles until the fatigue crack initiation in the experimentally tested 1st stage blade of the PZL-10W engine compressor with a geometric notch and tested in resonance conditions with amplitude 1.8 mm. The available scientific literature states that the number of cycles to initiation of a fatigue crack with a length of about 0.2 mm is 12.9 × 10^3^. The aforementioned value was used to evaluate the obtained numerical results. A group summary of all the results related to the various material configurations is presented in Table 6.

The comparative analysis shows that the closest to reality was the result for the Muralidharan-Manson fatigue model, with the Fatemi cyclic hardening. The obtained value was 92.88% of the experimental result. Given that, in an actual test, the crack was detected after expanding to 0.2 mm, it can be assumed that the actual durability was lower, which would make this result even closer to the actual result. In the case of the Manson hardening model, the best result was achieved in the Manson fatigue model (57.37%), but this result is slightly higher than half of the result from experimental tests. In the case of the Xianxin hardening model, the results are far from satisfactory (1 to 12.5% concerning the experimental result). In general, the results obtained using the cyclic hardening model of Fatemi are characterized by the greatest proximity to the experimental result, e.g., the result for the Median fatigue model was 133% of the experimental result, so it is too high by around 30%.

## 4. Conclusions

The presented work shows the results of an experiment aimed at determining the level of residual stresses on the compressor blade surfaces. The conducted tests showed that the highest values of compressive stresses are on the inner side of the blade (−713 MPa), close to the root of the blade. Interestingly, the lowest values of residual stresses were observed on the opposite side of the blade feather (−197 MPa). It was also observed that a stress relaxation of about 40% occurred as a result of the blade cutting.

Based on information on the residual stresses, as well as from literature, e.g., the state of stresses in a compressor blade with a geometric notch, vibrating with a resonance with an amplitude of 1.8 mm, it was possible to conduct an investigation, as a result of which, for several ε-N fatigue models and cyclic hardening models, it was possible to estimate the number of cycles until fatigue crack initiation.

The obtained results of the numerical fatigue analysis in conjunction with the literature data on the experimentally estimated fatigue life allowed for the quantitative and qualitative evaluation of the obtained results. Based on the conducted research, it was found that the best results were obtained using the cyclic hardening model according to Fatemi. At the same time, it was shown that acceptable results of numerical fatigue analysis were obtained for the Manson fatigue model.

The result closest to reality (experimental) was obtained by using the Muralidharan-Manson fatigue model and the Fatemi cyclic hardening model (92.88%). The Muralidharan-Manson fatigue model, in addition to the tensile strength (UTS) and Young’s modulus, also takes into account the percentage reduction from the static tensile test. Thanks to this, the model takes into account the plastic nature of a given material. Fatemi’s model does not impose a constant value of the cyclic hardening exponent (unlike the Manson model) but makes it dependent on the data from the fatigue model, hence a better fit to the material behavior is possible.

The obtained results show a clear tendency to underestimate the fatigue life using the Xianxin cyclic hardening model, while in the case of the Fatemi model, fatigue life may be overestimated (e.g., in cooperation with the Ong or Mitchell fatigue model).

In general, the investigation conducted allowed a quantitative and qualitative evaluation of popular material fatigue models and cyclic hardening models. The obtained results also indicate the necessity to conduct further research allowing the minimization of errors in the results of fatigue analyses for elements with a complex load condition and complex geometry.

## Figures and Tables

**Figure 1 materials-14-05279-f001:**
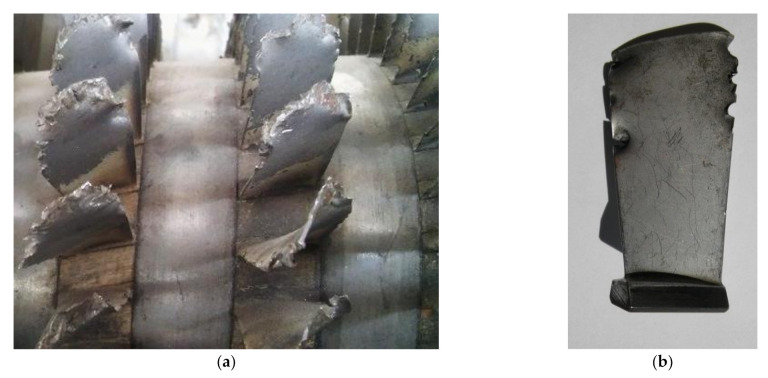
Photograph of a damaged compressor of a turbine engine (**a**) and a blade damaged by a collision with a foreign object (**b**).

**Figure 2 materials-14-05279-f002:**
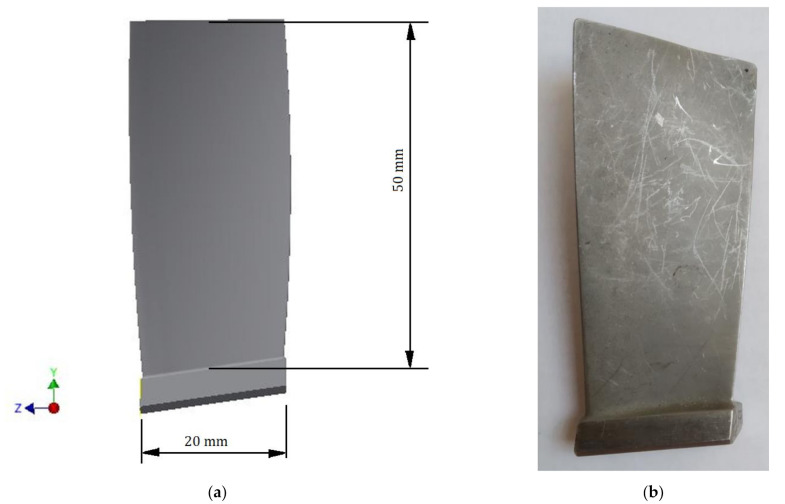
The geometrical model (**a**) of the examined blade (**b**).

**Figure 3 materials-14-05279-f003:**
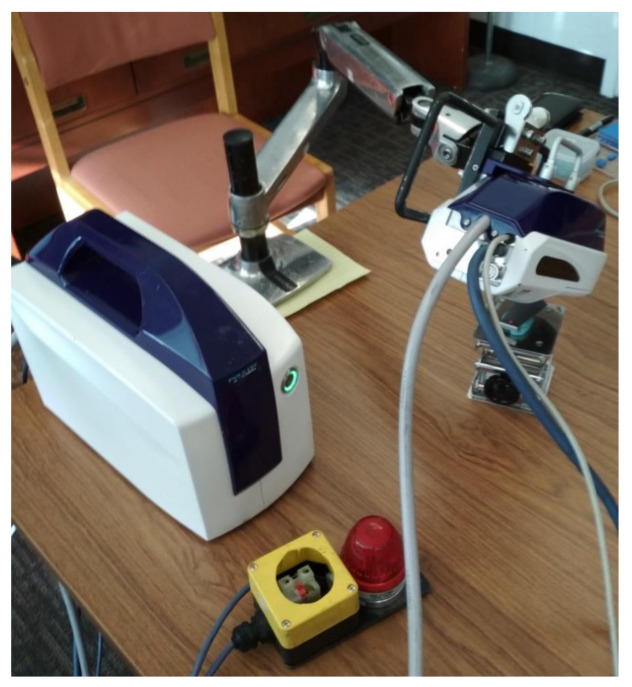
The experimental set-up for measuring the initial stresses on the blade surface.

**Figure 4 materials-14-05279-f004:**
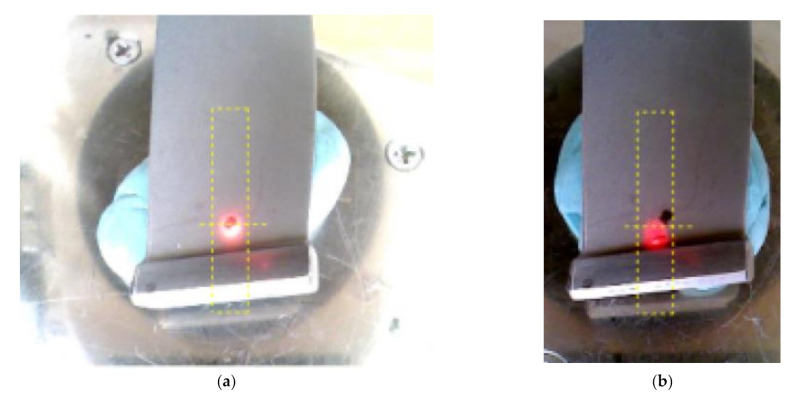
Measurement of residual/initial stresses on the blade surface, at a distance of h = 3 mm (**a**) and h = 1 mm (**b**) from the blade root.

**Figure 5 materials-14-05279-f005:**
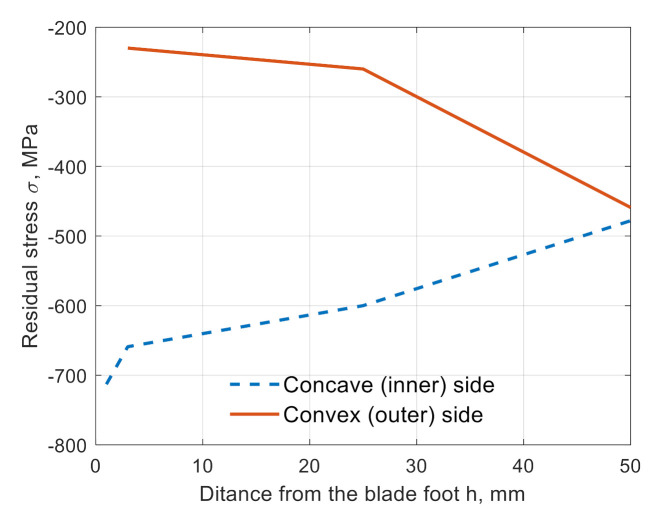
Distribution of initial stresses in the surface layer along with the height of the blade.

**Figure 6 materials-14-05279-f006:**
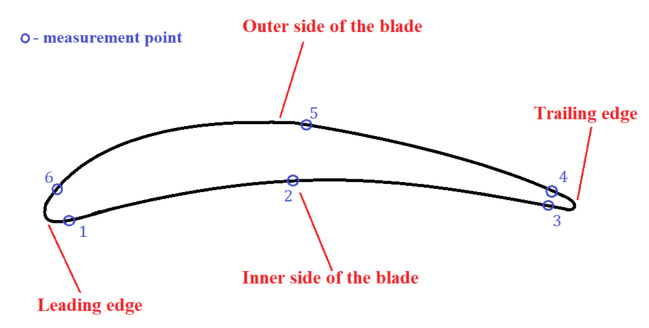
Location of residual stress measurement points along with the blade profile.

**Figure 7 materials-14-05279-f007:**
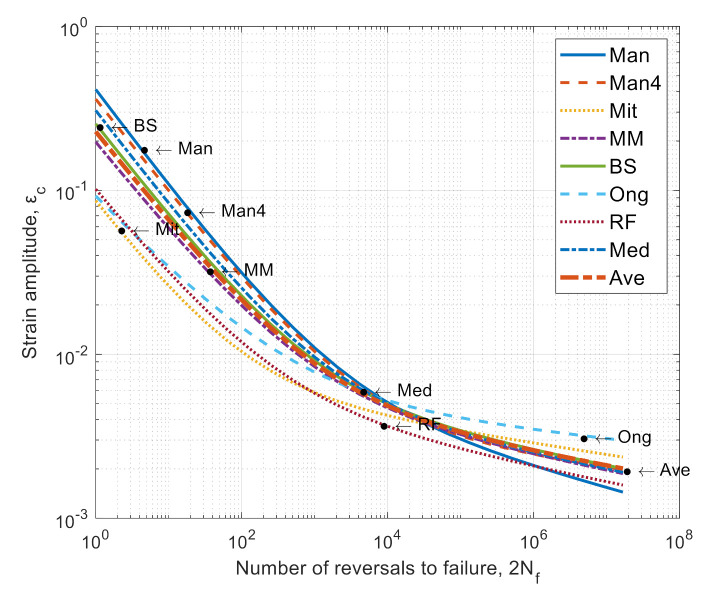
Summary of Manson-Coffin-Basquin curves for determined fatigue material models.

**Figure 8 materials-14-05279-f008:**
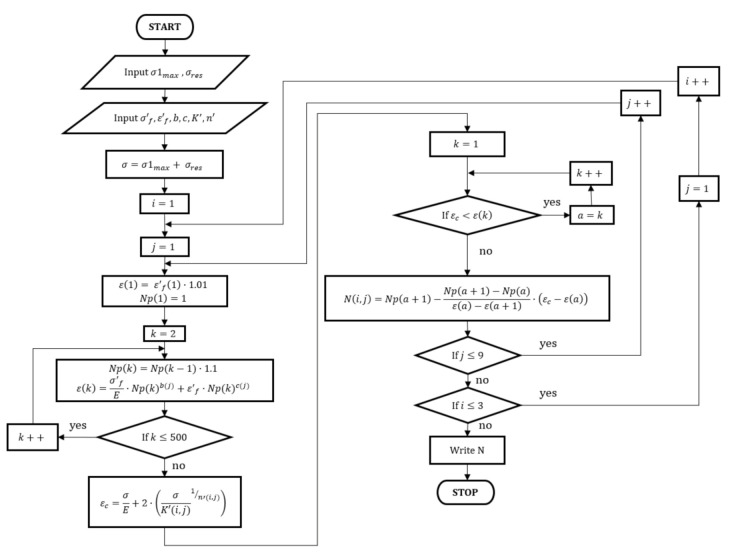
Scheme of the fatigue life assessment algorithm.

**Table 1 materials-14-05279-t001:** Results of the residual/initial stress measurements on the blade surface.

Distance from the Blade Root h, mm	Convex (Outer) Side of the Blade	Concave (Inner) Side of the Blade
-	Residual Stress σ_res_, MPa	
1	-	−713
3	−230	−659
25	−260	−600
50	−459	−478

**Table 2 materials-14-05279-t002:** Results of the residual stress in the cross-section located 3 mm above the blade root.

-	Inner (Concave) Side of the Blade			Outer (Convex) Side of the Blade		
Measurement point	1	2	3	4	5	6
Residual stress σ_res_, MPa	−478	−659	−255	−197	−230	−213

**Table 3 materials-14-05279-t003:** Material fatigue models estimated for EI-961 alloy, for ε-N analysis.

Name of the Model	Short-Name	σ ′_f_, MPa	ε ′_f_, mm/mm	b	c
Manson (1965)	Man	2280	0.61	−0.12	−0.6
4 point Manson (1965)	Man4	1172.2	0.528	−0.064	−0.57
Mitchell (1977)	Mit	1545	0.12	−0.068	−0.6
Muralidharan Manson (1988)	MM	1765.8	0.279	−0.09	−0.56
Baumel Seeger (1990)	BS	1800	0.368	−0.087	−0.58
Ong (1993)	Ong	1344	0.12	−0.047	−0.48
Roessle Fatemi (2000)	RF	1508.5	0.14	−0.09	−0.56
Median (2002)	Med	1800	0.45	−0.09	−0.59
Average	Ave	1651.9	0.327	−0.082	−0.57

**Table 4 materials-14-05279-t004:** Material cyclic hardening models estimated for EI-961 alloy.

-	K′_1_, MPa	n′_1_	K′_2_, MPa	n′_2_	K′_3_, MPa	n′_3_
Man	2516.9	0.2	2516.9	0.2	1775.5	0.17
Man4	1332		1258.3	0.11		
Mit	2361		1967.9	0.11		
MM	2279.8		2168.3	0.16		
BS	2198.4		2091.2	0.15		
Ong	2053.8		1656.1	0.10		
RF	2236.7		2070.2	0.16		
Med	2111.7		2033.2	0.15		
Ave	2136.3		1970.3	0.14		
	Manson′s model		Fatemi′s model		Xianxin′s model	

**Table 5 materials-14-05279-t005:** Results of numerical fatigue life tests for the compressor blade with a notch tested in resonance conditions.

-	Cyclic hardening model			
-	-	Manson′s	Fatemi′s	Xianxin′s
ε-N Fatigue material model	Man	$7.4\times 10^{3}$	$7.4\times 10^{3}$	$1.61\times 10^{3}$
	Man4	$0.03\times 10^{3}$	$1.08\times 10^{3}$	$1.47\times 10^{3}$
	Mit	$0.46\times 10^{3}$	$89.12\times 10^{3}$	$0.13\times 10^{3}$
	MM	$1.62\times 10^{3}$	$11.98\times 10^{3}$	$0.7\times 10^{3}$
	BS	$1.49\times 10^{3}$	$21.77\times 10^{3}$	$0.92\times 10^{3}$
	Ong	$0.31\times 10^{3}$	$401\times 10^{3}$	$0.37\times 10^{3}$
	RF	$0.37\times 10^{3}$	$1.96\times 10^{3}$	$0.2\times 10^{3}$
	Med	$1.22\times 10^{3}$	$17.17\times 10^{3}$	$1.11\times 10^{3}$

**Table 6 materials-14-05279-t006:** The results of the numerical fatigue analysis related to the actual blade life (geometrical V-shape notch, resonance A = 1.8 mm), determined experimentally N_in_ = 12.9 × 10^3^) [27].

-	Cyclic Hardening Model			
-	-	Manson′s	Fatemi′s	Xianxin′s
ε-N Fatigue material model	Man	57.37%	57.37%	12.51%
	Man4	0.22%	8.42%	11.37%
	Mit	3.54%	690.88%	1.01%
	MM	12.52%	92.86%	5.43%
	BS	11.52%	168.79%	7.13%
	Ong	2.38%	3108.91%	2.90%
	RF	2.87%	15.16%	1.52%
	Med	9.48%	133.07%	8.59%

## Data Availability

Data is contained within the article.

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
