# Peer review of "Numerical and Experimental Assessment of the Effect of Residual Stresses on the Fatigue Strength of an Aircraft Blade"

_materials, 2021, doi:10.3390/ma14185279_

Round 1

Reviewer 1 Report

ID: materials-1357782

Title: Numerical and experimental assessment of the effect of residual stresses on the fatigue strength of an aircraft blade

The goal of this paper is to comparison between results of numerical fatigue analysis and that of experimental analysis about the turbine engine compressor blade. For the experimental analysis, authors used X-ray diffraction to obtain the residual stresses. For the numerical analysis, several fatigue material models, such as Manson, Mitchel, Baumel-Seeger, Muralidharan-Manson, Ong, Roessle-Fatemi, and Median’s model, were considered.

This paper is quite interesting because of the importance of the engineering maintenance in the aircraft safety. These fatigue life of the blade is directly connected to the safety issues.

Minor:

Line 97: What does the GOST:13Kh12N2V2MF mean ?

Line196 : In Table 4, For n’1 and n’3, does it have the same value for all cases?

Author Response

Dear Reviewer,

In the beginning, I would like to thank you very much for the time spent reviewing our manuscript and providing suggestions on how to improve our publication.

All formatting errors and typos have been corrected.

Regarding the question for line 97:

GOST:13Kh12N2V2MF is the description (original name of the blade material). The PZL-10W engine was made based on Russian standards, that’s why the original name was given in this paper. Unfortunately, there is no information about equvalent alloys in European or American standards.

Regarding the question for line 196:

n'1 is connected to Manson model and it gives constant n’ (equal to 0,2). In the case of n’3 (connected to Xianxin model) –it is based on UTS, Young Modulus, and stress during fracture – these material data are the same for all cases of the material model, that’s why this quantity is also constant.

We have made additional changes within the paper to better explain all the issues addressed in all other reviews.

We believe, that the changes introduced into the manuscript and our specific answers cleared up any ambiguities and that our publication in this stage is ready for publication in the Materials journal.

Reviewer 2 Report

There are so many inaccuracies in this paper that it is difficult to list them all.
It would be difficult to revise the paper and still publish it.

Author Response

Dear Reviewer,

In the beginning, I would like to thank you for the time spent reviewing our manuscript. Thanks to the involvement of the other three reviewers, some changes were made to the content of the publication.

We hope that the changes introduced to the manuscript, as well as our specific answers, cleared up any ambiguities and that our publication in its current form is ready for publication in the Materials journal.

Reviewer 3 Report

This study focused on determining the fatigue life of a turbine engine compressor blade. For this, a numerical analysis of the ε-N fatigue life was performed based on several fatigue material models: Manson, Mitchell, Baumel-Seeger, Muralidharan-Manson, Ong, Roessle-Fatemi, and Median, and also on the three fatigue models. cyclic hardening: Manson, Xianxin, and Fatemi. The values obtained were compared with other scientific works, having high importance when applied to engineering maintenance and from the point of aircraft safety view.

Congratulations. The work is good, it is easy to read, however, there are some concerns about your work, which can be addressed, improving the work and your understanding. Please see the attachment.

Author Response

Dear Reviewer,

In the beginning, I would like to thank you very much for the time spent reviewing our manuscript and providing suggestions on how to improve our publication.

Regarding #1: The references were added in the first 3 paragraphs.

Regarding #2: All trademarks were added. In the entire paper, only Matlab and Pulstec were mentioned. Thank you for your valuable suggestion.

Regarding #3: There was our mistake – finally, there were 24 cases all together. The manuscript was modified - the correct number of analyzed cases was given.

Regarding #4: There was a typo in the paper (one extra decimal point was entered in the presented value in Table 5). A corresponding amendment has been made.

Regarding #5: The reasons for such dispersion in the results may be both undervalued fatigue models (which were developed for ferrous alloys and do not necessarily have to be correct in the case of alloys with specialized applications and different chemical composition). The work aimed to verify the possibility of using these models in the case of a specific alloy. The second reason may be the thickness of the plasticized zone due to peening and the intensity of this surface treatment. Another may be the complexity of the blade geometry and its arches, which may also affect the fatigue life. Unfortunately, we have no information on the method and parameters of surface treatment. The tested blades were obtained directly from the aviation industry, unfortunately, due to export control regulations, no detailed information was provided on the above-mentioned parameters.

We have made additional changes within the paper to better explain all the issues discussed by you and by other reviewers.

We hope that the changes introduced to the manuscript, as well as our specific answers, cleared up any ambiguities and that our publication in its current form is ready for publication in the Materials journal.

Reviewer 4 Report

The authors carried out an assessment of the effect of the residual stress on the fatigue strength of an aircraft blade. While the present research should be of interest to readers of this journal, the paper cannot be published in its present form, as the paper lacks some important technical details and in-depth discussions. The following issues must be addressed to enhance the quality of this paper.

  • Stress is a tensor, which has six independent components, but the authors treat the ‘residual/initial stress’ as if it is a scalar (e.g., table 1, table 2). This seems incorrect.
  • Does the X-ray experiment provide a full-field stress distribution (i.e., stress tensor at any point of the specimen)? Or, does it only provide the stress tensor at a few locations of the specimen?
  • What are the key procedures of the X-ray experiment? The set-up shown in Figure 3 does not provide much insight into how the experiment works. A clearer schematic illustration with a detailed explanation should be added to describe the experimental setup and procedure.
  • Does the numerical assessment involve a finite element calculation of the whole blade? Or, is it an evaluation of a material fatigue life model at one specific point of the blade?
  • The algorithm given in Figure 8 is difficult to understand because the involved equations are not explained at all in the paper.
  • The key mathematical equations of different fatigue material models should be given in Section 3 or in an appendix of the manuscript. The main similarities and differences between all these models should be discussed.
  • The key mathematical equations of different cyclic hardening models should be given in Section 3 or in an appendix of the manuscript. The main similarities and differences between all these models should be discussed.
  • Please discuss why the Muralidharan-Manson fatigue model gives the best result compared to the experimental result shown in Table 6.
  • Please discuss why the cyclic hardening model of Fatemi gives the best result compared to the experimental result shown in Table 6.
  • On Page 10, the authors state that “all 27 cases of material configurations are summarized in Table 5”, but in Table 5 only 24 data are given. Is ‘27’ a typo?

Author Response

Dear Reviewer,

In the beginning, we would like to thank you very much for the time spent reviewing our manuscript and providing suggestions on how to improve our publication.

Below, are the statements from your review, which are quotted and the answers do follow directly below them.

Statement: Stress is a tensor, which has six independent components, but the authors treat the ‘residual/initial stress’ as if it is a scalar (e.g., table 1, table 2). This seems incorrect.

Answer: The tool used to measure the residual stresses showed only one value of the stresses (it did not provide any components). According to the information from the representative of the instrument company, the indicated value is related to the direction normal to the tested surface. The authors assumed that the indicated value (related to the direction perpendicular to the blade surface) is sufficient and sufficiently represents the state of stresses in the surface layer of the blade material.

Statement: Does the X-ray experiment provide a full-field stress distribution (i.e., stress tensor at any point of the specimen)? Or, does it only provide the stress tensor at a few locations of the specimen?

Answer: As mentioned in the previous answer, the instrument used gives only one (representative) value of stress. Provided instrument allows us only to measure stress in a few locations on both sides of the blade (as explained in the paper).

Statement: What are the key procedures of the X-ray experiment? The set-up shown in Figure 3 does not provide much insight into how the experiment works. A clearer schematic illustration with a detailed explanation should be added to describe the experimental setup and procedure.

Answer: The specimen is located under the camera on a stand. The test was carried out by a representative of Pulstec and we were not able to take more detailed photos or collect more information. The device itself is portable and can be manipulated in any way to correctly position it with the tested element.

Statement: Does the numerical assessment involve a finite element calculation of the whole blade? Or, is it an evaluation of a material fatigue life model at one specific point of the blade?

Answer: The work uses the results of the numerical FEM analysis presented in the publication [30]. In this study, the analysis of the most loaded point in the blade was assumed and the analysis included the value of the main and reduced stresses (Von-Mises) in the bottom of the notch. As part of previous numerical and experimental studies, it was determined that this is where the fatigue crack initiation takes place (which was also taken from the earlier studies cited in the discussed analysis). The stress values taken from the previously discussed publications, in conjunction with the determined fatigue data, were used to estimate the fatigue life.

Statement: The algorithm given in Figure 8 is difficult to understand because the involved equations are not explained at all in the paper.

Answer: Information about equations used was added in chapter 3.1. The rest of the information in the algorithm is just a mathematical loop or loading or displaying data.

Statement: The key mathematical equations of different fatigue material models should be given in Section 3 or an appendix of the manuscript. The main similarities and differences between all these models should be discussed.

Answer: Formulas describing the ε-N fatigue models used (in Appendix A) have been added.

Statement: The key mathematical equations of different cyclic hardening models should be given in Section 3 or an appendix of the manuscript. The main similarities and differences between all these models should be discussed.

Answer: Formulas describing the cyclic hardening models used (in Appendix A) have been added.

Statement: Please discuss why the Muralidharan-Manson fatigue model gives the best result compared to the experimental result shown in Table 6.

Answer: The Muralidharan-Manson fatigue model, in addition to the tensile strength (UTS) and Young's modulus, also takes into account the percentage reduction from the static tensile test. Because of this approach, the model takes into account the plastic nature of a given material. In the conclusions, a mention of possible reasons for such a result was included in the manuscript.

Statement: Please discuss why the cyclic hardening model of Fatemi gives the best result compared to the experimental result shown in Table 6.

Answer: Fatemi's model does not impose a constant value of the cyclic hardening exponent but makes it dependent on the data from the fatigue model, hence a better fit to the material behavior is possible. In the conclusions, a mention of possible reasons for such a result was included in the manuscript.

Statement: On Page 10, the authors state that “all 27 cases of material configurations are summarized in Table 5”, but in Table 5 only 24 data are given. Is ‘27’ a typo?

Answer: It was our mistake – finally there were 24 cases. The text was modified - the correct number of analyzed cases was given.

We have made additional changes within the paper to better explain all the issues discussed.

We hope that the changes introduced to the manuscript, as well as our specific answers, cleared up any ambiguities and that our publication in its current form is ready for publication in the Materials journal.

Round 2

Reviewer 2 Report

There is a lack of discussion on the effect of residual stress on fatigue life. Please discuss the effect a little more.

Author Response

(The authors gave the same response as above.)

Reviewer 4 Report

The authors have attempted to enhance the quality of the manuscript, but I cannot recommend acceptance of this paper in its current form, because a clear presentation of the methodology is still lacking. Specifically, the following questions and concerns must be properly addressed:

(1) Definitions of all the variables (e.g., Nf, epsilon_C, E, sigma) in equations should be added right before or after the corresponding equation.

(2) A figure of stress-strain curves should be added to illustrate the definitions of the involved key material data (E, UTS, sigma_f, epsilon_f, RA). Particularly, the authors should explain how ‘RA’ is defined and how it is calculated from tensile test data.

(3) All the adopted values for the material data (E, UTS, sigma_f, epsilon_f, RA) should be provided before Table 3 is given. Also, please explain the source of these data. Are they experimentally measured by the authors or chosen from published literature?

(4) Most importantly, Figure 8 still looks quite confusing. The authors should give a clearer explanation of the algorithm.

Why do the authors calculate 500 components of Np(k) and epsilon(k) (k=1,2,3,…,500)? How is the factor 1.01 determined?

What is the physical motivation to compare the value of epsilon_c to epsilon(k) (k=1,2,3,…)?

When epsilon_c < epsilon(k) is not satisfied, N(i,j) is calculated from an equation that is not explained anywhere in the manuscript. What does this equation mean and how is it derived? In addition, what do the index i and j mean? Why do they range from 1~3 and 1~9, respectively?

Author Response

Dear Reviewer,

In the beginning, we would like to thank you very much for the time spent reviewing our manuscript and providing suggestions on how to improve our publication.

Below, are the statements from your review, which are quoted and the answers do follow directly below them.

Statement: (1) Definitions of all the variables (e.g., Nf, epsilon_C, E, sigma) in equations should be added right before or after the corresponding equation.

Answer: Definitions af all variables were added right after the corresponding equation (1, 2).

Statement: (2) A figure of stress-strain curves should be added to illustrate the definitions of the involved key material data (E, UTS, sigma_f, epsilon_f, RA). Particularly, the authors should explain how ‘RA’ is defined and how it is calculated from tensile test data.

Answer: Definitions and values af all variables were added in Appendix A. The values were taken from producer site [33] or from other literature papers. Due to the lack of access to the raw material (limited sales), it was not possible to perform a static tensile test. As part of the preparatory work, two tensile test specimens were cut from the blade profile, but due to the surface treatment and complex shape, these are not shown in this publication. The performed static tensile test was only used for the experimental verification of the literature data.

Statement: (3) All the adopted values for the material data (E, UTS, sigma_f, epsilon_f, RA) should be provided before Table 3 is given. Also, please explain the source of these data. Are they experimentally measured by the authors or chosen from published literature?

Answer: Due to the fact that specialized strength data are used in the calculations from Annex A, it was decided to include them together with the definitions in the said Annex. As mentioned earlier, the multiplicities are taken from the literature and confirmed experimentally on non-normative samples.

Statement: (4) Most importantly, Figure 8 still looks quite confusing. The authors should give a clearer explanation of the algorithm.

Answer:  The creation of the algorithm was not the purpose of the publication. The algorithm itself is only a tool used to calculate the fatigue life. The description of its operation has been extended before the algorithm.

Statement: Why do the authors calculate 500 components of Np(k) and epsilon(k) (k=1,2,3,…,500)? How is the factor 1.01 determined?

Answer: Factor 1.01 was determined experimentally. The value 500 is just a resolution of the assumed MCB curve for each material model. It was also experimentally checked for earlier assumed factor 1.01.

Statement: What is the physical motivation to compare the value of epsilon_c to epsilon(k) (k=1,2,3,…)?

Answer: Epsilon_C represents total strain for assumed residual stress. Epsilon(k) is a value from MCB plot. The algorithm looks for the case when epsilon_C is bigger than epsilon(k). If the condition is met, the algorithm estimates the fatigue life.

Statement: When epsilon_c < epsilon(k) is not satisfied, N(i,j) is calculated from an equation that is not explained anywhere in the manuscript. What does this equation mean and how is it derived? In addition, what do the index i and j mean? Why do they range from 1~3 and 1~9, respectively?

Answer: N (i, j) is used in the life calculation. When the algorithm finds the point where epsilon (c) is smaller than epsilon (k), then the next value in the life matrix of the MCB curve (for a given material) is subtracted from the next value in the life matrix (for a given material) and subtracts from it the "life portion", determined by Thales' theorem. The range from 1-3 and 1-9 performs the function of transition to the next calculation loop (with models of fatigue hardening and e-N fatigue models).

We have made additional changes within the paper to better explain all the issues discussed.

We hope that the changes introduced to the manuscript, as well as our specific answers, cleared up any ambiguities and that our publication in its current form is ready for publication in the Materials journal.